# Pain Improvement in Parkinson’s Disease Patients Treated with Safinamide: Results from the SAFINONMOTOR Study

**DOI:** 10.3390/jpm11080798

**Published:** 2021-08-16

**Authors:** Diego Santos García, Rosa Yáñez Baña, Carmen Labandeira Guerra, Maria Icíar Cimas Hernando, Iria Cabo López, Jose Manuel Paz González, Maria Gema Alonso Losada, Maria José Gonzalez Palmás, Carlos Cores Bartolomé, Cristina Martínez Miró

**Affiliations:** 1Department of Neurology, CHUAC, Complejo Hospitalario Universitario de A Coruña, 15006 A Coruña, Spain; jpazg1@hotmail.com (J.M.P.G.); Carlos.Cores.Bartolome@sergas.es (C.C.B.); Cristina.Martinez.Miro@sergas.es (C.M.M.); 2Department of Neurology, CHUO, Complejo Hospitalario Universitario de Ourense, 32005 Ourense, Spain; ryanezb@telefonica.net; 3Department of Neurology, CHUVI, Complejo Hospitalario Universitario de Vigo, 36213 Vigo, Spain; carmen.labandeira@hotmail.com (C.L.G.); gemavarita@gmail.com (M.G.A.L.); 4Hospital de Povisa, 36211 Vigo, Spain; icimash@mundo-r.com; 5Department of Neurology, CHOP, Complejo Hospitalario Universitario de Pontevedra, 36002 Pontevedra, Spain; icabol@yahoo.es (I.C.L.); Maria.Jose.Gonzalez.Palmas@sergas.es (M.J.G.P.)

**Keywords:** effectiveness, non-motor symptoms, pain, Parkinson’s disease, safinamide

## Abstract

Background and objective: Pain is a frequent and disabling symptom in Parkinson’s disease (PD) patients. Our aim was to analyze the effectiveness of safinamide on pain in PD patients from the SAFINONMOTOR (an open-label study of the effectiveness of SAFInamide on NON-MOTOR symptoms in Parkinson´s disease patients) study. Material and Methods: SAFINONMOTOR is a prospective open-label single-arm study conducted in five centers from Spain. In this analysis, a secondary objective of the study, the score in the KPPS (King´s Parkinson´s Disease Pain Scale) at V1 (baseline) and V4 (6 months ± 1 month) were compared. Wilcoxon´s rank sum test was performed to test the changes from V1 to V4. Results: Forty-four (88%) out of 50 PD patients (age 68.5 ± 9.12 years; 58% women; 6.4 ± 5.1 years from diagnosis) completed the study. The KPPS total score was reduced by 43.6% (from 40.04 ± 36.18 in V1 to 22.60 ± 21.42 in V4; *p* < 0.0001). By domains, improvement was observed in musculoskeletal (−35.9%; *p* = 0.009), fluctuation-related (−51.7%; *p* = 0.020), nocturnal (−46.1%; *p* = 0.001), discoloration and/or edema/swelling (−50.4%; *p* = 0.009) and radicular pain (−40.1%; *p* = 0.048). A total of 21 adverse events in 11 patients (22%) were reported, five being severe, but not related to safinamide. Conclusion: Safinamide is well tolerated and improves pain in PD patients at 6 months. Future studies are necessary to analyze the possible beneficial effect of safinamide on pain in PD patients.

## 1. Introduction

Parkinson’s disease (PD) is a chronic, progressive neurodegenerative disease characterized by the presence of motor symptoms and non-motor symptoms (NMS) [1]. Among these NMS, pain is observed in approximately 30%–50% of PD patients and the incidence can increase even to 68%–85% when all types of pain are taken into account [2]. Pain can appear at any time during the disease, and can be present before diagnosis [3]. Pain in PD is frequently under-recognized and is often inadequately treated [4]. As there are different types of pain, it is very important to identify what type of pain is present in the patient for trying to treat it [5]. Dopaminergic therapy may improve dystonic pain, musculoskeletal pain related to rigidity and akinesia, as well as akathisia in PD [6]. Moreover, not only dopamine but other neurotransmitters such as glutamate have been implicated in the modulation of pain in PD [7].

Safinamide is an oral α-aminoamide derivative marketed for the treatment of PD with both dopaminergic properties, namely highly selective and reversible inhibition of monoamine oxidase B, and non-dopamimetic properties, namely selective sodium channel blockade and calcium channel modulation, with consequent inhibition of excessive glutamate release [8]. In previous trials, safinamide improved both motor scores and duration of “on time”, and was also well-tolerated and safe [9,10]. Furthermore, data from some studies suggest a possible benefit of PD patients after treatment with safinamide in global NMS burden [11,12] and in some NMS in particular such as sleep, mood or urinary symptoms [13,14,15]. With regard to pain, very recently Geroin et al. [16] observed, after 3 months of add-on safinamide therapy, a significant improvement in the primary outcomes (KPPS (KPPS King’s Parkinson’s Disease Pain Scale), BPI (Brief Pain Inventory) Intensity and Interference and NRS (Numeric Rating Scale)) in 13 PD patients with pain. Post-hoc analysis of some trials suggest a possible effect of safinamide on pain in PD patients [17,18], and a recent systematic review and meta-analysis concluded that safinamide is an important adjunct to standard Parkinsonian medication for alleviating pain in PD [19]. Moreover, there is an on-going phase IV multi-center, randomized, double-blind, placebo controlled study about the effect of safinamide on pain in PD patients experiencing motor fluctuations and PD related chronic pain (https://clinicaltrials.gov/ct2/show/NCT03841604 (accessed on 13 August 2021); NCT03841604).

We observed very recently an improvement in the global NMS burden in 50 PD patients from the SAFINONMOTOR study [12]. In this analysis, a secondary objective of the SAFINONMOTOR study, we evaluated in detail the change in pain throughout the 6-month follow-up in PD patients treated with safinamide.

## 2. Material and Methods

SAFINONMOTOR is a multicenter, observational (phase IV), prospective, open-label, follow-up study conducted in five centers from Spain. Patients with PD according to the United Kingdom Parkinson’s Disease Society Brain Bank criteria [20] without dementia [21] who were considered for treating with safinamide by the neurologist having a NMSS total score ≥ 40 were included. SAFINONMOTOR methods are available in: https://www.ncbi.nlm.nih.gov/pmc/articles/PMC7999475/ (accessed on 13 August 2021) [12].

The study included 4 visits: V1 (baseline); V2 (1 months ± 7 days); V3 (3 months ± 15 days); V4 (6 months ± 15 days, end of the Observational Period). Subjects completed the KPPS [22] in all visits. The KPPS has been validated as the first specific rating scale to evaluate the burden of pain in the context of PD. The KPPS assesses seven different domains including 14 items corresponding to the diverse modalities of pain identified in PD. Domains 1 (musculoskeletal pain) and 2 (chronic pain) are nociceptive pain; neuropathic pain is included in domains 2 and 6 (discoloration; edema/swelling). Additionally, the scale includes fluctuation-related pain (domain 3), nocturnal pain (such as pain related to restless legs syndrome) (domain 4), orofacial pain (domain 5), and radicular pain (domain 7). Each item is scored by severity (0, none to 3, very severe) multiplied by frequency (0, never to 4, all the time) resulting in a sub-score of 0 to12, the sum of which gives the total score with a theoretical range from 0 to 168. Moreover, a visual analogue scale (VAS) was used for assessing pain (VAS-PAIN) [23] in each visit as well. Specifically, a straight horizontal line of 100 mm was used. The ends are defined as the extreme limits of the parameter to be measured orientated from the left (worst) to the right (best). Information on sociodemographic aspects, factors related to PD, comorbidity, and treatment was collected. Moreover, other scales were administered by protocol in different visits of the study. The methodology of the SAFINONMOTOR study can be consulted in https://www.mdpi.com/2076-3425/11/3/316/htm (accessed on 13 August 2021) [12]. Importantly, the analysis about the change in pain (KPPS total score) from V1 to V4 was a specific proposed secondary objective in the protocol of the SAFINONMOTOR study.

Safinamide was administered as once-daily 50 mg pill for 1 month and switched to 100 mg/day at V2 except at the discretion of neurologists (1 neurologist expert on PD in each center) according to the needs of the patients in their clinical practice. In the case of receiving rasagiline, the minimum washout period was 15 days. During follow-up, any other medications than safinamide should not have been modified (regimen, doses, etc.) except if the neurologist considered these changes absolutely necessary. All the changes in medications and levodopa equivalent daily dose (LEDD) [24] were recorded.

### 2.1. Data Analysis

Data were processed using SPSS 20.0 for Windows. Continuous variables were expressed as the mean ± SD or median and quartiles, depending on whether they were normally distributed. Relationships between variables were evaluated using the Mann-Whitney U test, Student´s t-test, and the, Pearson´s or Spearman´s correlation coefficient as appropriate (distribution for variables was verified by Kolmogorov-Smirnov test). The change from V1 to V4 in the KPPS total score was the principal efficacy outcome variable in this analysis. Moreover, the change in the score of each domain of the KPPS, VAS-PAIN score, and “Pain and discomfort” PDQ-39 score domain were analyzed. Analyses of efficacy variables were performed with the ITT data set (all subjects who receive at least 1 pill of safinamide and had a baseline and treatment observation for the primary efficacy outcome measure). A paired-sample t-test or Wilcoxon´s rank sum test, as appropriate, was performed to test the change from baseline.

In order to investigate which factors were predictors of improvement in pain in this cohort, linear regression models were used (KPPS total score change from V1 to V4 as dependent variable). Any variable with univariate associations with *p*-values < 0.20 were included in a multivariable model, and a backwards selection process was used to remove variables individually until all remaining variables were significant at the 0.10 level [12]. The variables considered for the analysis were: 1) at baseline (V1): age, gender, disease duration, LEDD, and KPPS total score; 2) at the end of the follow-up (V4): change from V1 to V4 in UPDRS-III, UPDRS-IV, FOGQ, NMSS, ESS, PSQI, BDI-II, VAFS-Physical, and VAFS-Mental. Tolerance and variance inflation factor (VIF) were used to detect multicollinearity (multicollinearity was considered problematic when tolerance was less than 0.2 and, simultaneously, the value of VIF 10 and above). A level of *p* < 0.05 was considered significant.

Finally, as was previously reported [12], adverse events were collected.

### 2.2. Standard Protocol Approvals, Registrations, and Patient Consents

Approval from the Comité de Ética de la Investigación Clínica de Galicia from Spain (2018-052; 28/FEB/2019) was obtained. All participants signed a consent form.

### 2.3. Data Availability

The data that support the findings of this study are available from the corresponding author upon reasonable request.

## 3. Results

Fifty PD patients were included in the study between May 2019 and February 2020 (age 68.5 ± 9.12 years; 58% females). Data on sociodemographic aspects, comorbidities, antiparkinsonian drugs and other therapies have been previously published [12] (Appendix A).

At 6 months, 44 patients completed the follow-up (88%) and a lower KPPS total score compared to baseline was observed in 28 out of 43 patients (65.1%), the same score in 6 patients (13.9%), and a higher score in 6 patients (13.9%). The KPPS total score was reduced from V1 to V4 by 43.6% (from 40.04 ± 36.18 in V1 to 22.6 ± 21.42 in V4; *p* < 0.0001) (Table 1 and Figure 1). Considering the different domains of the KPPS, a significant change between V1 and V4 was observed in the KPPS-domain 1 (musculoskeletal pain) (from 48.44 ± 39.74 to 31.06 ± 30.15; *p* = 0.009), KPPS-domain 3 (fluctuation-related pain) (from 11.11 ± 15.08 to 5.37 ± 10.74; *p* = 0.020), KPPS-domain 4 (nocturnal pain) (from 23.18 ± 27.30 to 12.50 ± 26.17; *p* = 0.001), KPPS-domain 6 (discoloration, edema/swelling) (from 11.46 ± 17.99 to 5.68 ± 11.65; *p* = 0.009), and KPPS-domain 7 (radicular pain) (from 16.32 ± 30.6 to 9.66 ± 22.15; *p* = 0.048) (Table 1). With regards to the VAS-PAIN score, there was a trend to significance reduction at V4 (from 4.61 ± 3.22 in V1 to 3.67 ± 2.69 in V2; *p* = 0.071) (Table 1). Compared to the score at V1, the change at V2 and V3 for the KPPS total score was also significant, but differences were not observed between the score from V2 and V3 to V4 (Figure 1 and Table 1). A significant decrease on “Pain and discomfort” PDQ-39 score domain was also observed at V4 (from 44.56 ± 27.35 to 33.33 ± 19.93; *p* = 0.018). Changes from V1 to V4 in different scales of the SAFINONMOTOR study have been previously published [12] (Appendix A).

Data are presented as box plots, with the box representing the median and the two middle quartiles (25%–75%). *p* values were computed using the Wilcoxon signed-rank test. Mild outliers (O) are data points that are more extreme than Q1 − 1.5 ∗ IQR or Q3 + 1.5 ∗ IQR.

KPPS-1—Musculoskeletal pain; KPPS-2—Chronic pain; KPPS-3—Fluctuation-related pain; KPPS-4—Nocturnal pain; KPPS-5—Oro-facial pain; KPPS-6—Discoloration, edema/swealing; KPPS-7—Radicular pain.

A moderate correlation was observed between the change from V1 to V4 in the KPPS total score and the change from V1 to V4 in the scores of NMSS (r = 0.577; *p* < 0.0001), VAS-PAIN (r = 0.438; *p* < 0.001), VAFS-Physical (r = 0.339; *p* = 0.028), PDQ-39SI (r = 0.326; *p* = 0.038), and PDQ-39SI-domain 8 (pain and discomfort) (r = 0.426; *p* < 0.0001). By domains, a strong correlation was observed between the change from V1 to V4 in the KPPS-domain 2 (chronic pain) and the change in the NMSS total score (r = 0.604; *p* < 0.0001). Other significant correlations between changes from V1 to V4 in the score of KPPS domains and the change from V1 to V4 in the score of the rest of scales are shown in Table 2. A strong correlation was observed between the change from V1 to V4 in the KPPS total score and the KPPS total score at baseline (r = −0.783; *p* < 0.0001), so the higher score at baseline, the greater the decrease in the score from V1 to V4.

In the multivariate analysis, a greater decrease in the NMSS total score from V1 to V4 (β = 0.242; 95%CI 0.022–0.402; *p* = 0.030) and a higher KPPS total score at baseline (β = 0.660; 95%CI 0.363–0.718; *p* < 0.0001) were associated with a greater decrease in the KPPS total score from V1 to V4 (adjusted R-squared 0.639; Durbin-Watson test = 2.428) (Table 3). In the final model (only these two variables), tolerance was 0.743 and VIF was 1.347.

When PD patients were divided in two groups, patients with a baseline KPPS total score < 50 (34/48 [70.8%]; mean age 68.21 ± 9.4; 52.9% females) and PD patients with a baseline KPPS total score ≥ 50 (14/48 [29.2%]; mean age 69.93 ± 8.6; 71.4% females), a significant difference was observed in the change from V1 to V4 in the KPPS total score (−5.31 ± 14.54 vs. −40.46 ± 38.81; *p* < 0.0001) (Figure 2). However, this was not observed for the VAS-PAIN (*p* = 0.175).

Data are presented as box plots, with the box representing the median and the two middle quartiles (25%–75%). *p* values were computed using the Wilcoxon signed-rank test.

KPPS, King´s PD Pain Scale; VAS-Pain, Visual Analog Scale-Pain.

Twenty-one adverse events in 16 patients (32%) were reported, 5 of them severe but not related to safinamide. This information has been reported before [12] (Appendix A). At baseline, 32% of the patients were taking any antidepressant agent, 38% benzodiazepines, 4% antipsychotics, and 22% analgesics (10% non-steroidal anti-inflammatory drugs; 10% opioids; 8% paracetamol; 8% antiepileptic drugs; Appendix A). During the follow-up, changes in treatment were made in 7 patients, but only in 5 patients were changes seen with drugs related to PD symptoms (Appendix A). Only one patient stopped with a medication (amlodipin/valsartan/hydrochlorothiazide) and no changes were reported with regard to analgesic drugs after 6-month follow-up.

## 4. Discussion

The present study observes that pain improved in patients with PD 6 months after starting with safinamide. Specifically, patients improved in musculoskeletal, fluctuation-related, nocturnal, discoloration and/or edema/swelling and radicular pain. Moreover, patients improved their health-related quality of life (QoL) and a correlation between QoL improvement and pain improvement was observed. Safinamide, as it has previously reported, was well tolerated.

Pain is a frequent and distressing symptom in PD occurring in 30%–85% of subjects (mean = 66%) [25]. There are various classification systems of pain for PD [10,26,27]. Ford´s pain classification from 2010 [10] remains the most-cited one and considered five types: musculoskeletal, radicular/neuropathic, dystonic, central or primary pain, and akathisia. Recently, Chaudhuri et al. [22] validated the King’s Parkinson´s Disease Pain Scale (KPPS), a new scale for assessing pain in PD patients. It includes different types of pain distributed in seven domains: nociceptive pain (musculoskeletal and chronic pain); neuropathic pain (chronic pain, discoloration and edema/swelling, and radicular pain); fluctuation-related pain; nocturnal pain; and orofacial pain. Importantly, different types of pain can be present at the same time in a patient and the management depends on it. To our best knowledge, only one 12-week, single-center, phase IV, prospective observational study analyzed the changes in pain in a cohort of PD patients receiving safinamide (100 mg/day) using the KPPS [16]. The results of the change in the KPPS total score were very similar to our finding (mean ± SD change from baseline of −19.3 ± 10.5; *p* < 0.05), but the sample was small (N = 13), and the numeric value of the KPPS total score at baseline was not provided (this seems to be about 40 points according to a figure), and, importantly, they information about the domains of the scale was not given. Although our study is not a double-blind trial like that of Cattaneo et al. [17], it is a prospective study in which a specific scale to measure the change in pain perception was used, the objective defined “a priori”, and not a post-analysis analysis.

Some types of pain can improve with dopaminergic medication, especially musculoskeletal and dystonic pain, akathisia or, in general, any pain emerging during the OFF state [2,6,10]. In our study, after 6 months with safinamide, pain as a whole and specifically five types of pain improved. With regard to this relation between pain and dopaminergic stimulus, we observed a significant correlation between the improvement in motor complications (UPDRS-IV) and chronic, fluctuation-related and nocturnal pain but unlike Geroin et al. [16] not with pain as a whole (KPPS total score). It suggests that reducing OFF time may improve pain, especially some types of pain. An example are nocturnal disturbances related to motor symptoms, including nocturnal akinesia, early-morning dystonia, painful cramps, tremor, and difficulty turning in bed [28]. Another interesting finding is the correlation observed between the global NMS burden improvement (change from V1 to V4 in the NMSS total score) and the improvement from V1 to V4 in pain as a whole (KPPS total score), and especially in chronic pain and fluctuation-related pain. It is well known that NMS are related in part to motor fluctuations and that reducing OFF time may improve NMS burden [29]. Moreover, NMS burden and chronic pain are related with depression and fatigue [30,31,32], and we observed a correlation between the improvement in chronic pain, fluctuation-related pain and/or nocturnal pain and the improvement in mood and/or physical fatigue after 6-month follow-up. However, in the linear model and after adjustment to baseline KPPS total score, the only predictor of pain improvement was the decrease in the global NMS burden.

Although the involvement of dopamine in the central modulation of pain is well established [33], other neurotransmitters, including glutamate, play an important role in pain signals, as suggested by the poor response of non-dystonic pain to levodopa [34]. Elevated glutamatergic neurotransmission is observed during neuropathic pain, and an imbalance between dopaminergic and non-dopaminergic systems might contribute to chronic pain in PD [35]. Moreover, deep brain stimulation targeted to the subthalamic nucleus was shown to reduce the glutamatergic overstimulation of the globus pallidus and improve the musculoskeletal and central neuropathic pain [36]. Safinamide is a drug for the treatment of PD with a double mechanism of action, dopaminergic and non-dopaminergic. Safinamide modulates the release of glutamate, and glutamate is believed to be involved in dyskinesia, chronic pain and mood disorders in PD patients [8]. It has been suggested that this second action mechanism could explain at least in part the favorable effect of safinamide in some NMS [11,13], but it is not clear whether the dopaminergic action of safinamide could be more potent than that of other MAO-B inhibitors such as rasagiline or if benefits could be related to its effect involving the glutamatergic system, or both [37,38]. We observed that the improvement in the NMS burden (NMSS total score) observed in 50 PD patients from the SAFINONMOTOR study 6 months after starting with safinamide with a mean dose of 99 mg/day did not significantly differ to the improvement observed at 1 month with a mean dose of 55 mg/day [12]. For pain, the same observation occurred when pain as a whole was considered, although fluctuation-related and discoloration/edema/swelling pain improved significantly at 6-month follow-up with a mean dose of 98.7 mg/day compared to 1-month follow-up with a mean dose of 53.8 mg/day.

The domain “Pain and discomfort” of the PDQ-39 improved at 6-month follow-up and a moderate correlation was observed between the improvement in pain (KPPS total score) and QoL (PDQ-39SI total score). In a post-hoc analysis, two of the three items of the PDQ-39 related to pain and the overall “Pain and discomfort” domain score improved with safinamide compared with placebo after 6 months [17] and even 2 years [18]. Moreover, safinamide reduced the concomitant use of analgesics of about 24% and 79.7% of pain reduction ascribed to safinamide was attributable to a direct effect of the drug [17]. However, in our study, after 6 months no patients taking at least one analgesic drug at baseline (22%) stopped this. The fact of trying not to modify, as far as possible, the rest of the treatments during the follow-up to avoid interpretation biases in the results could explain this.

The most important limitation of the present study is related to the study design itself and, since there is not a comparative placebo arm, the results should be interpreted very carefully. In line with this, a phase IV multicenter, randomized, double-blind, placebo controlled study about the effect of safinamide on pain in PD patients experiencing motor fluctuations and PD related chronic pain (https://clinicaltrials.gov/ct2/show/NCT03841604; (accessed on 13 August 2021) NCT03841604). Furthermore, the sample size is rather small and, for some variables, the information was not collected in all cases. The results are based on scales that collect the opinion of the patient and we did not include any objective method for assessing pain perception [33]. In fact, the improvement in mood could influence the perception of symptoms and the response in other scales [39]. Four visits were by phone and the effect that COVID-19 pandemic [40] could have in these patients is unclear. On the other hand, this is the first study in which changes on pain in PD patients receiving safinamide have been exhaustively analyzed.

In conclusion, safinamide is well tolerated and could improve pain in PD patients. Well-designed randomized double-blind trials are necessary to analyze in detail the effect of safinamide on pain. Especially interesting could be the analysis of pain with objective methods and with regards to action mechanism, dopaminergic and/or glutamatergic, in patients receiving safinamide.

## Figures and Tables

**Figure 1 jpm-11-00798-f001:**
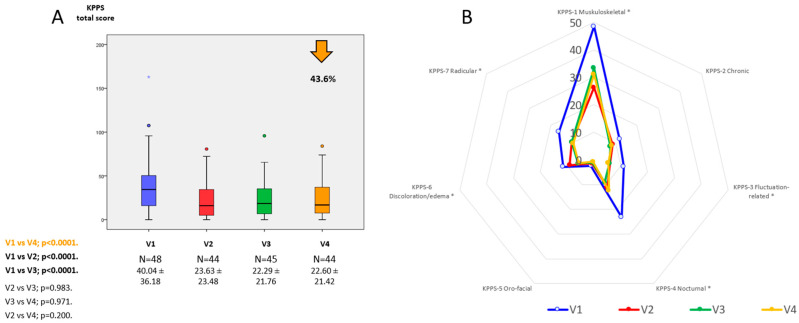
(**A**) KPPS total score at V1 (baseline), V2 (1 months ± 7 days), V3 (3 months ± 15 days), and V4 (6 months ± 15 days). The change at V2, V3, and V4 was significant compared to the score at V1 (*p* < 0.05 for all analysis; V4 vs. V1; V2 vs. V1; V3 vs. V1). *, *p* < 0.05. (**B**) Mean score on each domain of the KPPS scale at V1 (blue), V2 (red), V3 (green), and V4 (orange). The difference between V1 and V4 was significant for KPPS-domain 1 (musculoskeletal pain) (*p* = 0.009), KPPS-domain 3 (fluctuation-related pain) (*p* = 0.020), KPPS-domain 4 (nocturnal pain) (*p* = 0.001), KPPS-domain 6 (discoloration, edema/swelling) (*p* = 0.009), and KPPS-domain 7 (radicular pain) (*p* = 0.048).

**Figure 2 jpm-11-00798-f002:**
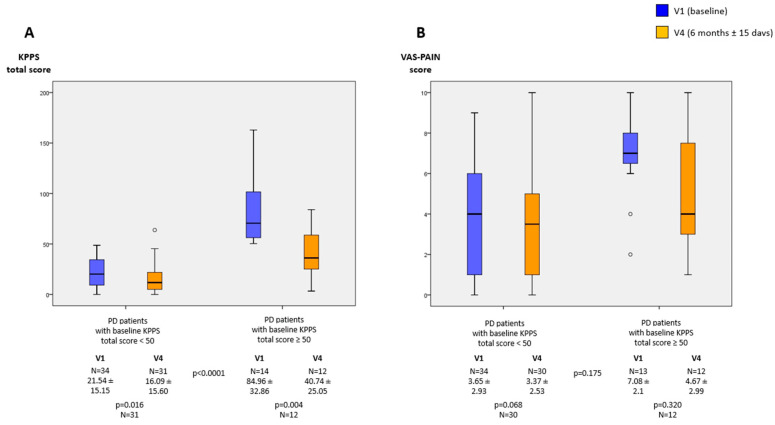
(**A**) KPPS total score at V1 (baseline) and V4 (6 months ± 15 days) with respect to the KPPS total score at baseline, <50 vs. ≥50. (**B**) VAS-PAIN score at V1 (baseline) and V4 (6 months ± 15 days) with respect to the KPPS total score at baseline, <50 vs. ≥50.

**Table 1 jpm-11-00798-t001:** Change in the total KPPS total score, its domains, VAS-PAIN total score and “Pain and discomfort” PDQ-39 score domain between the visits of the study: V1 (N = 50), V2 (N = 47), V3 (N = 45), V4 (N = 44).

	V1	V2	V3	V4	*p* ^a^	*p* ^b^	*p* ^c^	*p* ^d^
KPPS	40.04 ± 36.18	23.63 ± 23.48	22.29 ± 21.76	22.60 ± 21.42	<0.0001	<0.0001	<0.0001	0.2
- Musculo-skeletal pain	48.44 ± 39.74	26.32 ± 28.97	33.33 ± 31.38	31.06 ± 30.15	0.009	0.006	<0.0001	0.009
- Chronic pain	11.89 ± 21.14	8.8 ± 13.39	7.68 ± 15.04	8.24 ± 15.24	0.636	0.326	0.406	0.872
- Fluctuation-related pain	11.11 ± 15.08	5.99 ± 12.04	5.86 ± 13.35	5.37 ± 10.74	0.02	0.04	0.03	0.02
- Nocturnal pain	23.18 ± 27.30	11.17 ± 17.62	9.25 ± 14.91	12.50 ± 26.17	0.001	0.002	0.002	0.087
- Oro-facial pain	2.49 ± 9.58	1.51 ± 6.94	1.23 ± 3.86	0.82 ± 3.20	1	0.673	0.673	1
- Discoloration, edema/swelling	11.46 ± 17.99	9.99 ± 15.25	5.83 ± 11.56	5.68 ± 11.65	0.009	0.031	0.454	0.009
- Radicular pain	16.32 ± 30.60	10.22 ± 23.49	10.37 ± 22.76	9.66 ± 22.15	0.048	0.192	0.137	0.677
VAS-PAIN	4.61 ± 3.22	3.64 ± 2.73	3.67 ± 2.52	3.67 ± 2.69	0.071	0.047	0.078	0.071
PDQ-39SI Pain and discomfort	44.56 ± 27.35	36.87 ± 25.64	29.62 ± 22.58	33.33 ± 19.93	0.018	<0.0001	0.018	0.537
Dose of safinamide (mg/day)	N. A.	53.84 ± 13.49	96.15 ± 13.49	98.72 ± 8.00				

*p* values were computed using the Wilcoxon signed-rank test. The results represent mean ± SD. Domains of the NMSS were expressed as a percentage to be able to establish comparisons of severity between; *p* ^a^, V4 vs. V1; *p* ^b^, V3 vs. V1; *p* ^c^, V2 vs. V1; *p* ^d^, V4 vs. V2. N. A., Not applicable. KPPD, King´s PD Pain Scale; VAS-Pain, Visual Analog Scale-Pain.

**Table 2 jpm-11-00798-t002:** Correlations between the changes in KPPS (KPPS total score and KPPS domains) and changes in othes scales from V1 (baseline) to V4 (6 years ± 15 days).

	KPPS	KPPS-1	KPPS-2	KPPS-3	KPPS-4	KPPS-5	KPPS-6	KPPS-7
	Total Score	Musculoskel.	Chronic	Fluctuation-Related	Nocturnal	Oro-Facial	Disc./Edema	Radicular
MOTOR ASSESSMENT								
UPDRS-III-ON	0.255	0.182	0.245	0.165	−0.092	0.126	0.216	0.147
UPDRS-IV	0.031	−0.010	0.377 ***	0.385 ***	0.370 ***	0.11	−0.124	−0.217
FOGQ	0.106	−0.061	0.21	0.343 ***	−0.135	0.172	0.094	−0.280
NON MOTOR ASSESSMENT								
NMSS total score	0.577 *	0.188	0.604 *	0.520 *	0.283	0.281	0.399 ***	−0.129
ESS	0.118	0.026	−0.045	0.321 ***	0.01	−0.110	0.095	−0.008
PSQI	0.062	−0.201	0.187	0.238	0.065	−0.034	0.113	−0.270
BDI-II	0.05	0.109	0.478 *	0.354 ***	0.29	0.242	0.345 ***	0.098
VAS-PAIN	0.438 *	0.245	0.191	0.176	0.407 ***	0.285	0.286	0.111
VAFS-Physical	0.339 ***	0.1	0.182	0.089	0.489 **	0.272	0.104	0.024
VAFS-Mental	0.206	−0.085	0.191	0.181	0.296	0.082	0.009	0.045
QOL AND AUTONOMY								
PDQ-39SI	0.326 ***	0.072	0.247	0.439 ***	0.069	0.055	0.315 ***	0.019
- Mobility	0.166	−0.187	0.285	0.335 ***	−0.061	0.004	0.243	−0.070
- Activities of daily living	0.196	0.008	0.147	0.263	0.034	0.009	0.361 ***	−0.224
- Emotional well-being	0.252	0.241	0.211	0.229	0.027	0.069	0.135	−0.026
- Stigmatization	−0.147	−0.154	−0.117	0.035	−0.020	−0.153	−0.187	−0.023
- Social support	0.006	0.064	−0.340 ***	0.035	0.165	0.092	0.053	−0.073
- Cognition	0.183	0.29	−0.035	0.029	0.26	−0.021	−0.014	0.207
- Communication	0.15	0.325 ***	−0.033	0.235	−0.049	0.097	−0.051	0.149
- Pain and discomfort	0.426 *	0.238	0.188	0.232	0.275	0.294	0.369 ***	0.207
ADLS	0.005	0.011	0.016	−0.095	−0.054	−0.050	0.025	0.276

Spearman correlation test were applied. *, *p* < 0.0001; **, *p* < 0.001; ***, *p* < 0.05. In bold are expressed significant values. ADLS, Schwab & England Activities of Daily Living Scale; BDI, Beck Depression Inventory; ESS, Epworth Sleepiness Scale; FOGQ, Freezing Of Gait Questionnaire, H&Y: Hoenh & Yahr; KPPS, King´s PD Pain Scale; NMSS, Non-Motor Symptoms Scale; NPI, Neuropsychiatric Inventory; PDQ-39SI, 39-item Parkinson’s Disease Quality of Life Questionnaire Summary Index; PSQI, Pittsburgh Sleep Quality Index; UPDRS, Unified Parkinson’s Disease Rating Scale; VAFS, Visual Analog Fatigue Scale; VAS-Pain, Visual Analog Scale-Pain.

**Table 3 jpm-11-00798-t003:** Linear regression model for factors associated with pain improvement after 6-month follow-up (change in the KPPS total score from V1 to V4).

	β ^a^	β ^b^	95% IC ^a^	95% IC ^b^	*p* ^a^	*p* ^b^
At V1 (baseline)						
Age	0.041	N. A.	−0.848–1.102	N. A.	0.794	N. A.
Gender	0.326	N. A.	1.592–35.156	N. A.	0.033	N. A.
Disease duration	0.166	N. A.	0.019–1.499	N. A.	0.293	N. A.
LEDD	0.359	N. A.	0.003–0.037	N. A.	0.02	N. A.
KPPS total score	0.783	0.66	0.480–0.802	0.363–0.718	<0.0001	<0.0001
Change at V4 (from V1 to V4)						
UPDRS-III	0.255	N. A.	−0.237–1.698	N. A.	0.134	N. A.
UPDRS-IV	0.031	N. A.	−3.984–4.710	N. A.	0.866	N. A.
FOGQ	0.106	N. A.	−1.147–2.268	N. A.	0.511	N. A.
NMSS	0.577	0.242	0.280–0.731	0.022–0.402	<0.0001	0.03
ESS	0.118	N. A.	−1.233–2.695	N. A.	0.457	N. A.
PSQI	0.062	N. A.	−0.369–0.215	N. A.	0.711	N. A.
BDI-II	0.508	N. A.	0.895–2.949	N. A.	0.001	N. A.
VAFS-Physical	0.339	N. A.	0.328–5.511	N. A.	0.028	N. A.
VAFS-Mental	0.206	N. A.	−0.957–4.629	N. A.	0.192	N. A.

Dependent variable: change from V1 to V4 in the KPPS total score. β standardized coefficient and 95% IC are shown. ^a^, univariate analysis; ^b^, multivariate analysis (Durbin-Watson test = 2.428; R^2^ = 0.639). N. A., not applicable. ADLS, Schwab & England Activities of Daily Living Scale; BDI, Beck Depression Inventory; ESS, Epworth Sleepiness Scale; FOGQ, Freezing Of Gait Questionnaire, H&Y: Hoenh & Yahr; KPPS, King´s PD Pain Scale; NMSS, Non-Motor Symptoms Scale; NPI, Neuropsychiatric Inventory; PDQ-39SI, 39-item Parkinson’s Disease Quality of Life Questionnaire Summary Index; PSQI, Pittsburgh Sleep Quality Index; UPDRS, Unified Parkinson’s Disease Rating Scale; VAFS, Visual Analog Fatigue Scale; VAS-Pain, Visual Analog Scale-Pain.

## Data Availability

The protocol and the statistical analysis plan are available on request. Deidentified participant data are not available for legal and ethical reasons.

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
