# Peer review of "Pain Improvement in Parkinson’s Disease Patients Treated with Safinamide: Results from the SAFINONMOTOR Study"

_jpm, 2021, doi:10.3390/jpm11080798_

Round 1

Reviewer 1 Report

In this manuscript, the authors provide a more detailed analysis of their previously published dataset (reference 12) of 50 patients in an open label, Industry sponsored trial for with some of the visits were conducted by phone.

The data are presented and discussed adequately, the discussion is appropriate and the relevant literature is cited. Yet, I have some concerns:

The only novel findings are in Table 3. For the clinician, the most important information is that the reduction of pain (KPPS total score) significantly contributes to patients’ quality of life (PDQ39 total score). Yet, this can also be seen from the data in reference 12 and Table 1. The other correlations in Table 3 are reassuring but not too helpful, e.g. the correlation between the reduction of fluctuation related pain (D3) with the reduction in dyskinesias (UPDRS IV).

Table 1 is the same as in reference 12. This is possible because both are MPDI journals, but this reviewer does not consider it adequate. Figure 1 does not provide data that is not in Table 1. The information in Table 2 and Figure 2 are not helpful for the clinician.

This reviewer therefore recommends that the authors convert their manuscript into a short communication about the main finding (Table), omitting information that was already included in the first manuscript. If they choose not to do so, it would improve readability of Table 3 to include the labels “D1” etc. for the domains of the KPPS into Tables 1 and 2 and into Figure 1B.

While we have to accept that de-identified participant data is not available, this reviewer considers this aspect no longer appropriate and encourages the authors to include this possibility in further studies to increase their possibility to be published.

Author Response

REVIEWER 1:

In this manuscript, the authors provide a more detailed analysis of their previously published dataset (reference 12) of 50 patients in an open label, Industry sponsored trial for with some of the visits were conducted by phone.

The data are presented and discussed adequately, the discussion is appropriate and the relevant literature is cited. Yet, I have some concerns:

The only novel findings are in Table 3. For the clinician, the most important information is that the reduction of pain (KPPS total score) significantly contributes to patients’ quality of life (PDQ39 total score). Yet, this can also be seen from the data in reference 12 and Table 1. The other correlations in Table 3 are reassuring but not too helpful, e.g. the correlation between the reduction of fluctuation related pain (D3) with the reduction in dyskinesias (UPDRS IV).

AUTHORS – Thank you very much for your comment. As we indicated in the manuscript, this is an analysis of a secondary objective of the project SAFINONMOTOR. We are very clear on this and we reference the previous study published in Brain Sciences and indicate that it is a secondary objective initially designed before starting the study. The SAFINONMOTOR is a study with many evaluations that involved a significant effort of resources, and we believe that the information is very useful. In the first manuscript previously published in Brain Sciences, only data about KPPS at baseline and at 6 months was provided but here there is an extensive analysis about the data in the KPPS total score and its domains at baseline, 1 month, 3 months and 6 months, with specific data in tables and figures. Furthermore, we compare the results between visits and with different doses of safinamide in the total score and each domain. Moreover, correlations between the changes in the KPPS (KPPS total score and KPPS domains) and changes in other scales from V1 (baseline) to V4 (6 years ± 15 days) have been analyzed. The manuscript is focused on pain. The only study published was conducted in only 13 patients and they didn´t showed information about the domains of the scale. Here, we collected all the information and this information we believe that is something novel. Importantly, previous information published in Brain Science from this study appears here only such as supplementary material except Table 1. Both Figures are novel again. We agree with you and Table 1 has been now considered as Supplementary Material, thanks. Moreover, a linear analysis for determining independent variables associated with pain improvement in this cohort has been added. In summary, we believe that this manuscript is focused on pain and provide new relevant information. If you review the introduction, aim, results, analysis, discussion, and conclusion, it is totally different from the first previous referenced study and in general, a very complete study.

Table 1 is the same as in reference 12. This is possible because both are MPDI journals, but this reviewer does not consider it adequate. Figure 1 does not provide data that is not in Table 1. The information in Table 2 and Figure 2 are not helpful for the clinician.

This reviewer therefore recommends that the authors convert their manuscript into a short communication about the main finding (Table), omitting information that was already included in the first manuscript. If they choose not to do so, it would improve readability of Table 3 to include the labels “D1” etc. for the domains of the KPPS into Tables 1 and 2 and into Figure 1B.

AUTHORS – Thank you very much for your comment. We agree with you that the information in table 1 has been previously published, so we change this aspect and Table 1 is presented as Supplementary Table with a reference to the previous study. Moreover, we include the changes in PDQ-39SI – domain 9 (Pain and discomfort) as data in Table 2 (Table 1 in R1 version). Figure 1 provides information on the changes in the main variable and its domains in a visual way that helps to understand the observed effect. With regard to Figure 2, we think that it is interesting because it provides information about a novel sub-analysis with clinical implications: a possible better effect in PD patients with greater global pain perception. As you suggest, we have improved the readibility of domains in Tables and Figures.

While we have to accept that de-identified participant data is not available, this reviewer considers this aspect no longer appropriate and encourages the authors to include this possibility in further studies to increase their possibility to be published.

AUTHORS – Thank you very much for your comment. We are transparent and all the publications derived from the SAFINONMOTOR study are clear in their objective and results based on previous publications. The sentence has been changed: “The data that support the findings of this study are available from the corresponding author upon reasonable request”.

REVIEWER 1:

In this manuscript, the authors provide a more detailed analysis of their previously published dataset (reference 12) of 50 patients in an open label, Industry sponsored trial for with some of the visits were conducted by phone.

The data are presented and discussed adequately, the discussion is appropriate and the relevant literature is cited. Yet, I have some concerns:

The only novel findings are in Table 3. For the clinician, the most important information is that the reduction of pain (KPPS total score) significantly contributes to patients’ quality of life (PDQ39 total score). Yet, this can also be seen from the data in reference 12 and Table 1. The other correlations in Table 3 are reassuring but not too helpful, e.g. the correlation between the reduction of fluctuation related pain (D3) with the reduction in dyskinesias (UPDRS IV).

AUTHORS – Thank you very much for your comment. As we indicated in the manuscript, this is an analysis of a secondary objective of the project SAFINONMOTOR. We are very clear on this and we reference the previous study published in Brain Sciences and indicate that it is a secondary objective initially designed before starting the study. The SAFINONMOTOR is a study with many evaluations that involved a significant effort of resources, and we believe that the information is very useful. In the first manuscript previously published in Brain Sciences, only data about KPPS at baseline and at 6 months was provided but here there is an extensive analysis about the data in the KPPS total score and its domains at baseline, 1 month, 3 months and 6 months, with specific data in tables and figures. Furthermore, we compare the results between visits and with different doses of safinamide in the total score and each domain. Moreover, correlations between the changes in the KPPS (KPPS total score and KPPS domains) and changes in other scales from V1 (baseline) to V4 (6 years ± 15 days) have been analyzed. The manuscript is focused on pain. The only study published was conducted in only 13 patients and they didn´t showed information about the domains of the scale. Here, we collected all the information and this information we believe that is something novel. Importantly, previous information published in Brain Science from this study appears here only such as supplementary material except Table 1. Both Figures are novel again. We agree with you and Table 1 has been now considered as Supplementary Material, thanks. Moreover, a linear analysis for determining independent variables associated with pain improvement in this cohort has been added. In summary, we believe that this manuscript is focused on pain and provide new relevant information. If you review the introduction, aim, results, analysis, discussion, and conclusion, it is totally different from the first previous referenced study and in general, a very complete study.

Table 1 is the same as in reference 12. This is possible because both are MPDI journals, but this reviewer does not consider it adequate. Figure 1 does not provide data that is not in Table 1. The information in Table 2 and Figure 2 are not helpful for the clinician.

This reviewer therefore recommends that the authors convert their manuscript into a short communication about the main finding (Table), omitting information that was already included in the first manuscript. If they choose not to do so, it would improve readability of Table 3 to include the labels “D1” etc. for the domains of the KPPS into Tables 1 and 2 and into Figure 1B.

AUTHORS – Thank you very much for your comment. We agree with you that the information in table 1 has been previously published, so we change this aspect and Table 1 is presented as Supplementary Table with a reference to the previous study. Moreover, we include the changes in PDQ-39SI – domain 9 (Pain and discomfort) as data in Table 2 (Table 1 in R1 version). Figure 1 provides information on the changes in the main variable and its domains in a visual way that helps to understand the observed effect. With regard to Figure 2, we think that it is interesting because it provides information about a novel sub-analysis with clinical implications: a possible better effect in PD patients with greater global pain perception. As you suggest, we have improved the readibility of domains in Tables and Figures.

While we have to accept that de-identified participant data is not available, this reviewer considers this aspect no longer appropriate and encourages the authors to include this possibility in further studies to increase their possibility to be published.

AUTHORS – Thank you very much for your comment. We are transparent and all the publications derived from the SAFINONMOTOR study are clear in their objective and results based on previous publications. The sentence has been changed: “The data that support the findings of this study are available from the corresponding author upon reasonable request”.

Reviewer 2 Report

In the present study, the authors have investigated the efficacy of Safinamide in improving NMS symptoms, primarily different kinds of pain in PD patients. Their conclusion is that safinamide could improve pain in PD patients. The manuscript is well written but the English needs some work. Introduction is good, methods have good details, results are presented nicely and discussion included shortcomings of previous studies, potential mechanisms of the drugs and caveats of the present study.

I have a few concerns and would like to see them getting addressed before recommending publication.

1) How did the authors decide to use 50 mg/day dosage when 100 mg/day has already been used safely in reference 17?

2) Could they elaborate on how their study is different from 17 and what new are they showing in this one. The potential of safinamide as a pain medication for PD has already been shown in 17 before.

3) Could they also write down what were the “severe” side effects and how did they rule out that they were not related to the drug without a placebo drug?

4) I would have liked to see a placebo control and double-blind trials, in the absence of which it becomes hard to conclude these results with full confidence.

5) Was there any gender bias observed in the efficiency of the drug as pain reducer in PD patients?

Author Response

REVIEWER 2:

In the present study, the authors have investigated the efficacy of Safinamide in improving NMS symptoms, primarily different kinds of pain in PD patients. Their conclusion is that safinamide could improve pain in PD patients. The manuscript is well written but the English needs some work. Introduction is good, methods have good details, results are presented nicely and discussion included shortcomings of previous studies, potential mechanisms of the drugs and caveats of the present study.

AUTHORS – Thank you very much for your comment. English style has reviewed by a native English teacher.

I have a few concerns and would like to see them getting addressed before recommending publication.

1) How did the authors decide to use 50 mg/day dosage when 100 mg/day has already been used safely in reference 17?

AUTHORS – Thank you very much for your comment. As it is explained in Methods, “safinamide was administered as once-daily 50 mg pill for 1 month and switched to 100 mg/day at V2 except at the discretion of neurologists (1 neurologist expert on PD in each center) according to the needs of the patients in their clinical practice”. We decided this design with the intention of compare the effect with 50 mg vs 100 mg / day. In general, we didn´t observe differences in the effect between 50 and 100 mg. However, the correct interpretation is that there was a significant improvement with 50 mg 1 month after introducing the drug, which continued to be observed 5 months later, with 100 mg, since they are different evolutionary times.

2) Could they elaborate on how their study is different from 17 and what new are they showing in this one. The potential of safinamide as a pain medication for PD has already been shown in 17 before.

AUTHORS – Thank you very much for your comment. If you don't mind, you can read above the response to reviewer 1 regarding what this study brings compared to the previous one published in Brain Sciences (reference 12). Compared to the Cattaneo et al. study, this is a prospective study in which a specific scale for assessing pain was used, and to evaluate the effect of safinamide on pain was designed “a priori”, being not a post-hoc analysis. We add a commentary in the discussion: “Although our study is not a double-blind trial like Cattaneo et al. study [17], it is a prospective study in which a specific scale to measure the change in pain perception was used, being the objective defined “a priori”, and not a post-analysis analysis”.

3) Could they also write down what were the “severe” side effects and how did they rule out that they were not related to the drug without a placebo drug?

AUTHORS – Thank you very much for your comment. Severe adverse events were shown in Table 2.SM (Table 3 Supplementary Material in R1 version): urinary infection (N=1), pneumonia (n=1); respiratory insufficiency (n=1); dizziness (n=1); deep brain stimulation surgery (n=1; it was really not an severe adverse event but it was classified as it because the patient was hospitalized). The relationship or not with the drug was based on the opinion of the expert neurologist according to his criteria (symptoms, chronological relationship, comorbidities, other factors, etc.).

4) I would have liked to see a placebo control and double-blind trials, in the absence of which it becomes hard to conclude these results with full confidence.

AUTHORS – Thank you very much for your comment. We agree with you and of course it is necessary. This limitation is commented in discussion and a double-blind trial on-going is referenced: https://clinicaltrials.gov/ct2/show/NCT03841604; NCT03841604. In discussion, we talk about the possibility of improvement and the necessity of double-blind trials: “In conclusion, safinamide is well tolerated and could improve pain in PD patients. Well-designed randomized double-blind trials are necessary to analyze in detail the effect of safinamide on pain”. Importantly, we change the title of the manuscript: “Pain Improvement in Parkinson´s Disease Patients Treated with Safinamide: Results from the SAFINONMOTOR Study”. Thanks again.

5) Was there any gender bias observed in the efficiency of the drug as pain reducer in PD patients?

AUTHORS – Thank you very much for your comment. The reduction in the KPPS total score from V1 to V4 was significantly higher in females (from 48.3 ± 39.74 to 25.96 ± 21.5) than in males (from 25.2 ± 21.27 to 20.33 ± 21.85) (p=0.033). However, there is an important confusion factor, the KPPS total score at baseline, being higher in females than males (p=0.028). This is relevant because a higher KPPS total score at baseline was associated with a greater improvement (Figure 2). In relation to your comment, we have added a linear regression analysis to know what factors were predictors of improvement of pain (dependent variable: change from V1 to V4 in the KPPS total score). Thank you very much again. We believe that this completes the work by providing interesting information.

Round 2

Reviewer 2 Report

Thanks to the authors for making these changes and replying to my concerns. The revised version looks much better and I don't have other problem with it.